# Seven-Month Analysis of Five SARS-CoV-2 Antibody Assay Results after ChAdOx1 nCoV-19 Vaccination: Significant Decrease in SARS-CoV-2 Antibody Titer

**DOI:** 10.3390/diagnostics12010085

**Published:** 2021-12-30

**Authors:** Seri Jeong, Nuri Lee, Su-Kyung Lee, Eun-Jung Cho, Jungwon Hyun, Min-Jeong Park, Wonkeun Song, Eun-Ju Jung, Heungjeong Woo, Yu-Bin Seo, Jin-Ju Park, Hyun-Soo Kim

**Affiliations:** 1Department of Laboratory Medicine, Kangnam Sacred Heart Hospital, Hallym University College of Medicine, Seoul 07441, Korea; hehebox73@hallym.or.kr (S.J.); nurilee822@hallym.or.kr (N.L.); mjpark@hallym.or.kr (M.-J.P.); swonkeun@hallym.or.kr (W.S.); 2Department of Laboratory Medicine, Dongtan Sacred Heart Hospital, Hallym University College of Medicine, Hwaseong 18450, Korea; sklee1217@naver.com (S.-K.L.); ejlovi@hallym.or.kr (E.-J.C.); jungwonhyun@hallym.or.kr (J.H.); 3Division of Infectious Diseases, Department of Internal Medicine, Dongtan Sacred Heart Hospital, Hallym University College of Medicine, Hwaseong 18450, Korea; christer38@hallym.or.kr (E.-J.J.); infwoo@hallym.or.kr (H.W.); 4Division of Infectious Diseases, Department of Internal Medicine, Kangnam Sacred Heart Hospital, Hallym University College of Medicine, Seoul 07441, Korea; niceday@hallym.or.kr (Y.-B.S.); pjjhob@hallym.or.kr (J.-J.P.)

**Keywords:** SARS-CoV-2, antibodies, assay, ChAdOx1 nCoV-19, vaccine, viral load

## Abstract

We investigated the longevity rates of antibodies to severe acute respiratory syndrome coronavirus 2 (SARS-CoV-2) after a complete ChAdOx1 nCoV-19 vaccination, which are rare and important to estimate their efficacy and establish a vaccination strategy. We assessed the positivity rates and changes of titers before (T0) and at one month (T1), four months (T2), and seven months (T3) after a ChAdOx1 nCoV-19 vaccination using five SARS-CoV-2 antibody assays. A total of 874 serum samples were obtained from 228 (T0 and T1), 218 (T2), and 200 (T3) healthcare workers. The positive rates for all five assays were 0.0–0.9% at T0, 66.2–92.5% at T1, 98.2–100.0% at T2, and 66.0–100.0% at T3. The positive rates at T3 were decreased compared to those at T2. The median antibody titers of all the assays at T3 were significantly decreased compared to those at T2 (860.5 to 232.0 U/mL for Roche total, 1041.5 to 325.5 AU/mL for Abbott IgG, 10.9 to 2.3 index for Siemens IgG, 99.5% to 94.7% for SD Biosensor V1, and 88.5% to 38.2% for GenScript). A third-dose scheme can be considered based on our data generated from five representative assays. Our findings contribute insights into SARS-CoV-2 antibody assays and appropriate vaccination strategies.

## 1. Introduction

Since the coronavirus disease (COVID-19) pandemic caused by severe acute respiratory syndrome coronavirus 2 (SARS-CoV-2) infection continues to rage, universal vaccination has been approved as the most effective tool for limiting viral circulation and reducing the risk of poor outcomes [1,2,3]. To date, several vaccines have been developed, including those using viral proteins such as SARS-CoV-2 spike protein, DNA- and mRNA-based vaccines, and lipid-based mRNA nanoparticle vaccines [1]. The vaccines that have been administered most often are BNT162b2 (Pfizer, Inc., Collegeville, PA, USA), mRNA-1273 (Moderna, Inc., Cambridge, MA, USA), and ChAdOx1 nCoV-19 (AstraZeneca., Lund, Sweden) vaccines [4,5,6]. 

As SARS-CoV-2 infection continues, questions have recently been raised about the durability of these vaccines and the length of time for which SARS-CoV-2 immunogenicity lasts [7,8]. The maintenance of vaccine immunogenicity after a long period of time is important in determining whether additional booster vaccines are needed and in the selection of groups requiring booster vaccinations in case of limited vaccine supply. Numerous studies have been conducted on vaccine efficacy for the prevention of COVID-19 and the relationship of vaccines to neutralizing antibodies, but research on long-term durability of various vaccines have been limited so far. Few studies on the mRNA-1273 vaccine have been reported, showing a vaccine efficacy of 93.2% with a median follow-up period of 5.3 months [9] and a significant decrease in antibodies six months after the first injection [10]. Regarding the ChAdOx1 nCoV-19 vaccine, the only published data on longevity revealed that the vaccine is highly efficacious in the first 90 days after vaccination [11]. In addition, most studies use a single method to measure vaccine efficacy and have limitations in the standardized and accurate assessment of the SARS-CoV-2 immunity. Previous investigations have shown that there would be differences in the results of antibody titer according to each different antibody measurement method, reagent manufacturer, and target antigen [12,13]. 

Therefore, in this study, we intended to investigate vaccine efficacy utilizing quantitative antibody titers of SARS-CoV-2 seven months after the first injection of a complete ChAdOx1 nCoV-19 vaccination. The antibody titer was measured using five representative antibody reagents from various manufacturers and neutralization tests to evaluate the long-term durability of the vaccine without bias. In addition, the agreement and correlation between antibody assays were studied, and these insights may influence the adoption of the assays in various laboratory settings.

## 2. Materials and Methods

### 2.1. Study Population and Sample Collection 

A total of 200 healthcare workers from two university hospitals (Hallym University Dongtan Sacred Heart Hospital and Hallym University Kangnam Sacred Heart Hospital) who met the inclusion criteria were registered in this study. The main inclusion criteria were as follows: older than 20 years of age, eligible for vaccination, and upon provision of informed consent, including the acknowledgement of the purpose and design of this study. Serum samples were drawn from the participants to verify the SARS-CoV-2 antibody status before their first vaccinations (median: 2 days; T0); the first dose was administered between 4 and 12 March 2021. The second sampling event was carried out between 11 and 28 days later (T1). The second vaccination occurred between 20 May and 15 June 2021, and the third lot of sampling was conducted between 10 and 38 days after this, corresponding to 101–117 days after the first injection (T2). The fourth sampling event was executed 7 months (207–222 days) after the first dose, corresponding to 4 months (112–145 days) after the second vaccination (T3). A total of 228 samples were collected at T0 and T1. After T1, 10 healthcare workers were excluded due to their resignation, refusal of the second vaccination dose or blood sampling, or injection of the Pfizer-BioNTech vaccine. After T2, three participants resigned from their jobs, and 18 workers did not submit samples. In total, 200 samples were retained, aliquoted and assessed within 10 days. Before the experiments, the serum samples were stored at −70 °C in deep freezers. The results for T1 and T2 used in our previous research [2,3] were deposited to a public database (HARVARD Dataverse: https://doi.org/10.7910/DVN/HNDD7L (accessed on 25 November 2021); https://doi.org/10.7910/DVN/HPPSBA (accessed on 25 November 2021)) and extracted for this study.

### 2.2. Measurement of SARS-CoV-2 Antibodies

The determination of the serologic responses was performed using five SARS-CoV-2 antibody assays. First, an Elecsys Anti-SARS-CoV-2 S assay on the Elecsys Cobas e801 platform (Roche Diagnostics, Mannheim, Germany) that targeted the receptor-binding domain (RBD) was used to measure total antibodies based on an electrochemiluminescence immunoassay method (double-antigen sandwich principle). The cutoff for the Roche assay was 0.8 U/mL. The predefined master curve was adapted to a analyzer using the relevant calibration materials. They were standardized against the internal Roche standard for Elecsys Anti-SARS-CoV-2 S. This standard comprised an equimolar mixture of two monoclonal antibodies binding the spike-1 RBD at 2 different epitopes. One nanomolar of these antibodies corresponded to 20 U/mL of this assay. Controls for the various concentration ranges were run individually at least once every 24 hours. The values obtained were within the defined limits. Second, SARS-CoV-2 IgG II Quant on Alinity I (Abbott, Abbott Park, IL, USA) that used a chemiluminescent microparticle immunoassay to detect IgG antibodies targeting the RBD; its cutoff was 50 AU/mL. A SARS-CoV-2 IgG II Quant Calibrator Kit was used for the calibration when determining the anti-SARS-CoV-2 IgG antibody in sera. The kit contained anti-SARS-CoV-2 IgG in a phosphate buffer with a bovine stabilizer. Six different concentrations from 0.0 to 1666.7 AU/mL were included. A SARS-CoV-2 IgG II Quant Control Kit was utilized to estimate the test precision and detect the analytical deviation. The target concentration of the negative control was 2.3 AU/mL, whereas those of the positive controls were 166.0 and 602.5 Au/mL. Third, the SARS-CoV-2 IgG assay on the Atellica IM platform (Siemens, Munich, Germany) measured IgG antibodies targeting the RBD based on a chemiluminescence immunoassay method. Its cutoff was 1.0 index. After entering the master curve and the test definition, calibration was performed using the calibrators provided in each kit. The interpretation of the test results was based on the index value established with the calibrators. One negative control and two positive control materials were utilized for quality control procedures. Fourth, a STANDARD E SARS-CoV-2 nAb ELISA kit (SD Biosensor, Suwon, Korea) determined RBD-binding neutralizing antibodies, and its cutoff was 30% (percent inhibition (PI) value). The SD Biosensor STANDARD E SARS-CoV-2 nAb ELISA was composed of V1 and V2 assays targeting the Wuhan/UK variant and South Africa/Brazil variant, respectively. At least one positivity of V1 and V2 assay was considered a positive result for the SD Biosensor assay. We calculated the PI value using the following equation:PI value = (1 − (O.D. of sample/mean of O.D. of negative control)) × 100,(1)
where O.D. is the optical density. If the mean absorbance for the negative control was over 1.0 and the PI value of the positive control was over 80, the values were determined as acceptable. Fifth, a cPass SARS-CoV-2 neutralization antibody detection kit (GenScript, NJ, USA) utilized competitive ELISA and also measured the RBD binding neutralization antibodies with a 30% cutoff value. The operator determined the result of the sample by comparing the inhibition rate derived from the following equation: Percent Signal Inhibition = (1 − (O.D. of sample/mean of O.D. of negative control)) × 100.(2)

For quality control, the O.D. of the negative control was over 1.0, and that of the positive control was below 0.3. In addition, the coefficients of variation of the O.D. signal values for all replicates of the positive and negative samples should be within 10% in the same run. Both the GenScript cPASS SARS-CoV-2 neutralization antibody detection kit and the SD Biosensor STANDARD E SARS-CoV-2 nAb ELISA kit were surrogate virus neutralization tests, which could detect neutralizing antibodies blocking the interaction between the RBD and angiotensin-converting enzyme 2 coated on the ELISA plate. The ELISA procedures for the SD Biosensor and GenScript assays using an Epoch Microplate Spectrophotometer (BioTek Instruments, Winooski, VT, USA) and an ELx50 Filter Microplate Washer (BioTek Instruments) were similar to those in previous studies [2,3]. All experiments were performed in accordance with the manufacturer’s instructions. The coded samples for maintaining anonymity were examined in a single-blinded manner by laboratory technicians and scientific researchers.

### 2.3. Statistical Analysis 

Chi-square tests for nominal values and Mann–Whitney U tests or Kruskal–Wallis tests for continuous values were applied for assessment. The Steel-Dwass-Critchlow-Finger test for multiple comparisons was used to compare changes in antibody levels among T0–T3. The agreements between assays were evaluated according to Cohen’s kappa values using the similar categories in previous studies [2,3]. Briefly, Cohen’s kappa values of 0.61–0.80 and 0.81–1.00 were interpreted as substantial and almost perfect, respectively. Spearman’s rank correlation coefficients were applied to the assessment of correlations, as described in previous reports [2,3]. The Spearman’s rank correlation coefficients of 0.40–0.69 were moderate, and those of 0.70–0.89 were strong. The values above 0.90 were designated as very strong. Analyse-it Method Evaluation Edition software, version 2.26 (Analyse-it Software Ltd., Leeds, UK) and MedCalc software, version 19.8 (MedCalc Software Ltd., Ostend, Belgium) were utilized for analysis.

## 3. Results

### 3.1. Characteristics of Participants and Samples 

A total of 200 serum samples were collected from participants seven months after their first injection. The basic characteristics and serological responses are summarized in Table 1. The median age of the participants was 35.0 years and ranged from 22.0 to 60.0 years. The median of the sampling days after the first dose (T3) was 213.0 days (the 1st to 3rd quartile range: 210.0–215.0 days). This corresponded to a median of the sampling days after the second dose of 134.0 days (the 1st to 3rd quartile range: 132.0–137.0 days). Our participants comprised nurses (65.0%), laboratory technicians (27.0%), and doctors (7.0%). 

### 3.2. Qualitative Results of the SARS-CoV-2 Antibody Assays after Seven Months

The positivity rates for the SARS-CoV-2 antibody assays at T0–T3 are presented in Table 1. The positive rates seven months after the first injection for all the assays were decreased (97.0% for Abbott IgG, 86.0% for Siemens IgG, 98.0% for SD Biosensor V1, 92.0% for SD Biosensor V2, and 66.0% for the GenScript assay) when compared to the rates after the second dose, except for Roche total (100.0%). None of the studied characteristics, including sex, age, occupation, and hospital, were significantly related to the positivity of the five SARS-CoV-2 antibody assays seven months from the first injection. Age showed a decreasing probability of positivity, although not statistically significant (*p* = 0.667; Appendix A).

### 3.3. Quantitative Antibody Results of the SARS-CoV-2 Antibody Assays after Seven Months

Seven months from the first vaccination, the quantitative antibody levels decreased significantly for all assays (*p* < 0.001; Table 2 and Figure 1). The median values of Roche total, Abbott IgG, Siemens IgG, SD Biosensor V1, SD Biosensor V2, and GenScript in the samples after seven months reduced from 860.5 U/mL at T2 to 232.0 U/mL at T3, 1041.5 to 325.5 AU/mL, 10.9 to 2.3 index, 99.5% to 94.7%, 97.4% to 72.8%, and 88.5% to 38.2%, respectively (Table 2). The median values of the differences between the sampling times are presented in Table 2. One sample from Roche total, two from SD Biosensor V2, and two from GenScript at T3 showed increased titers when compared to those at T2. However, the differences were not significant (*p* = 0.749 for Roche total, *p* = 0.866 for SD Biosensor V2, and *p* = 0.866 for GenScript). At T3, the titers from Abbott IgG (*p* = 0.5292), Siemens IgG (*p* = 0.8047), and GenScript (*p* = 0.9996) were closer to those obtained at T1 than those observed at T2 (Figure 1). Despite the decreased levels, the median values of all the assays at T3 were higher than the cutoff values after a complete vaccination scheme. No participants were reported to be infected by SARS-CoV-2.

### 3.4. Agreement and Correlation between the SARS-CoV-2 Assays

The agreement rates between the results obtained from the five assays are presented in Table 3. The total agreement rates ranged from 85.6% (95% confidence interval (CI): 83.1%–87.9%) to 98.3% (95% CI: 97.2%–99.0%). Consistent with the results at T1 and T2, the rate between Abbott IgG and SD Biosensor neutralizing antibodies showed the highest agreement. The neutralizing and non-neutralizing antibody assays did not reveal significant differences. The kappa values ranged from 0.69 to 0.96, showing substantial or almost perfect agreements among the included assays. Abbott IgG and SD Biosensor neutralizing antibodies revealed the highest kappa value (0.96; 95% CI: 0.94–0.98) among all the assays. In contrast, the kappa values for GenScript with other assays were relatively lower. 

The correlations among the assays according to the sampling time after injection are summarized in Table 4. Abbott IgG and Siemens IgG showed the highest correlation (rho value = 0.976) among the assays. Meanwhile, the correlation between Roche total and Siemens IgG (rho value = 0.876) was lower than those of the others. The correlation graphs between Roche total and the other assays, such as Abbott IgG, Siemens IgG, SD Biosensor, and GenScript, showed two linear correlation patterns differentiating T1 from T2 and T3 (Appendix A), because of the weaker binding affinity at T1.

## 4. Discussion

In this study, we investigated the antibody longevity at seven months after the first injection of a two-dose ChAdOx1 nCoV-19 vaccination scheme for healthcare workers using five SARS-CoV-2 assays. The assays included one assay for total antibodies, two assays for IgG, and two surrogate virus neutralizing antibody assays. The positive rates after seven months ranged from 66.0% to 100.0% according to the type of assays. The median antibody levels of all assays at T3 were decreased compared to those at T2. 

The data available on the longevity of the antibody responses after SARS-CoV-2 vaccinations are very limited, especially for ChAdOx1 nCoV-19 (Table 5). There are few published data for mRNA vaccines, such as BNT162b2 and mRNA-1273, which were the first to obtain authorization in Europe and the United States [4]. A study determining IgG antibody levels 69 days after a two-dose BNT162b2 vaccination among healthcare professionals emphasizes the need for prospective immunosurveillance research [5]. A decline in antibody titers at three months after two doses of BNT162b2 in non-immunocompromised adults was reported [6]. The median titer measured by SARS-CoV-2 IgG II Quant from Abbott significantly dropped from 9356 AU/mL at 1.5 months to 3952 AU/mL at 3 months. Although the titers were low, our data also revealed a significant decline between T2 (1 month; 1041.5 AU/mL) and T3 (4 months; 325.5 AU/mL). Another study for antibody titers after a BNT162b2 vaccination at three months (1262 U/mL) presented a significant decline when compared to the peak response at one month (2204 U/mL) [7]. Although all participants still had detectable antibodies determined by Elecsys anti-SARS-CoV-2 spike total quantitative ECLIA from Roche Diagnostic with a cutoff of 0.8 U/mL, the decreasing trend of titers supports the necessity of an immunosurveillance study with longer follow-up. Our data showed a decline from 860.5 U/mL at one month to 232.0 U/mL at four months using the Roche total assay. The lowered titers in our study likely stemmed from the type of the injected vaccines. 

With respect to mRNA-1273 reporting a 93.2% vaccine efficacy in preventing COVID-19, more than five months of longevity was observed [13]. Another study investigating the durability of antibodies after an mRNA-1273 vaccination reported persistent binding and functional antibodies for six months. However, the report suggesting a faster decay of antibodies to viral variants urged additional booster vaccinations [11]. Our study also revealed a decreased positivity of antibodies to variants (SD Biosensor V1 = 98.0% versus SD Biosensor V2 = 92.0%) at seven months. There have been additional studies demonstrating the durability of antibodies after an mRNA-1273 vaccination [9,10,12]. Persistent binding and neutralizing antibodies at 119 days (*n* = 34 healthy participants) [12] or 209 days (*n* = 33 healthy individuals) [9] declining over time have been reported. A study that performed IgG analysis in 201 healthcare workers who were fully vaccinated with mRNA-1273 also showed peak levels at three months and a significant decline at six months after the first dose, similar to our data. These data support the introduction of a third dose.

Regarding the ChAdOx1n CoV-19 vaccine, minimal waning with high antibody levels at three months after a single dose has been reported [14,15]. However, to the best of our knowledge, no reports dealing with the longevity of antibodies after the full dose of ChAdOx1n CoV-19 vaccination at more than six months has been undertaken. A model of the decay of the neutralization titer over the seven months after vaccination predicted a significant loss in the protection from SARS-CoV-2 infection [16]. When considering this prediction model and the relatively low quantitative titer after a ChAdOx1 nCoV-19 vaccine compared to after an mRNA vaccine, the significant decline observed in our data at seven months was concordant with our anticipated results.

Older age was reported to be associated with lower rates of antibody responses. In the study on antibodies at three months after a BNT162b2 vaccination in healthcare workers, older participants had higher rates of decline [8]. In contrast, mRNA-1273 was effective in preventing the SARS-CoV-2 infection, regardless of participants being more than 65 years of age. Another study showed that antibodies after an mRNA-1273 vaccination in participants aged 56 to 70 and 71 and older were as potent and durable as those in individuals aged 18 to 55 [9,17]. Further, participants aged over 71 years retained neutralizing antibodies against the variants at six months after the second dose of mRNA-1273 [11]. In addition, the analysis of the effect of the age on the decrease in antibodies detected between three and six months after an mRNA-1273 vaccination did not reveal any influence [10]. In our study, older age seemed to be associated with a decrease in SARS-CoV-2 antibodies at seven months after a ChAdOx1 nCoV-19 vaccination; however, this was not statistically significant. According to a previous study on the ChAdOx1 nCoV-19 vaccine, the IgG and neutralizing antibody titers after the second dose were similar across age groups [18]. Further studies, including increased numbers of participants aged over 60 years with long-term follow-ups, are necessary for demonstrating antibody responses in older patients.

The representative SARS-CoV-2 antibody assay included in this study presented substantial or almost perfect agreement (kappa values = 0.69–0.96) and strong or very strong correlations with each other (rho value = 0.876–0.976) and were concordant with previous studies [19,20]. The strong correlation between antibodies against the spike protein and neutralization antibodies has been consistently reported. A previous study investigating the correlation between a quantitative anti-SARS-CoV-2 IgG assay and neutralization activity demonstrated a strong relationship (Spearman’s correlation coefficient = 0.819) in samples obtained during the COVID-19 pandemic [21]. In this study, Abbott IgG and SD Biosensor neutralizing antibody assays had the highest kappa value (0.96), and the agreement between the two was 98.3%, similar to that of assays (93.5%) in a previous study comparing 12 commercial SARS-CoV-2 immunoassays [22]. Meanwhile, the Cohen’s kappa values of Roche total and Abbott IgG with GenScript were 0.74 and 0.61, respectively, in a study evaluating immunoassays for SARS-CoV-2 antibodies [23]. The relatively low values were similar to our data (0.71 for Roche total and 0.69 for Abbott IgG with GenScript). In a previous study [24], the Spearman correlation analysis showed a weak correlation between an anti-SARS-CoV-2 antibody detected by commercially available serologic assays and the neutralizing activity in the samples from COVID-19 patients. There have been discrepancies in the relationship between assays measuring the anti-SARS-CoV-2 antibody and those determining the neutralizing antibody. The differences in the samples (from COVID-19 patients or participants after vaccination) and the utilized serologic assays (targeting spike or nucleocapsid protein) could be the causes of these discrepant results. In addition, the methods used to measure the neutralizing activity, such as surrogate virus-neutralizing tests, plaque reduction neutralization tests, and focus reduction neutralization assays, should be considered. Further studies including live-virus tests, such as plaque reduction neutralization tests and focus reduction neutralization assays, and sufficient samples obtained from diverse sources are necessary. Regarding the higher correlation of Abbott IgG and Siemens IgG (rho value = 0.976) compared to that of Roche total and Siemens IgG (rho value = 0.876), the measurement of antibody types and principles such as the double-antigen sandwich assay format detecting antibodies with high affinity could be the cause of this phenomenon. The two linear correlation patterns of Roche total could be observed because of the long-term follow-up of this study at one month, four months, and seven months. Owing to the affinity maturation, the binding strength of antibodies increases over time following infection or vaccination [25,26,27]. Therefore, the correlation patterns of Roche total differed at T1 from at T2 and T3. The characteristics of each assay, such as antibody detection with high affinity based on a double-antigen sandwich assay, higher sensitivity and specificity, and the agreement and correlation with neutralization assays, should be considered. The application of assays to clinical laboratories depends on the priority of each laboratory after the accumulation of sufficient data demonstrating the usefulness of the assays.

## 5. Conclusions

A full dose of ChAdOx1 nCoV-19 vaccination-induced antibody activities persisted for seven months after the first dose based on five representative SARS-CoV-2 antibody assays. However, the median antibody titers of these assays were significantly decreased at seven months in comparison with those at four months. The degree of decay in the antibody titers varied according to the types of the assays used. The agreements and correlations among the included assays were substantial and strong; however, the results should be interpreted cautiously considering the characteristics of all the assays and their respective cutoff values. To the best of our knowledge, we are the first to provide reliable data on serological responses seven months after the first injection of a full dose of ChAdOx1 nCoV-19 vaccination based on five representative SARS-CoV-2 antibody assays, including neutralization antibody assays. These results contribute to vaccination strategies by providing insight into anti-SARS-CoV-2 immunoassays in the fight against the SARS-CoV-2 infection. The characteristics of assays meeting the needs of each laboratory should be considered. Our findings also provide evidence that introducing a booster dose may be beneficial.

## Figures and Tables

**Figure 1 diagnostics-12-00085-f001:**
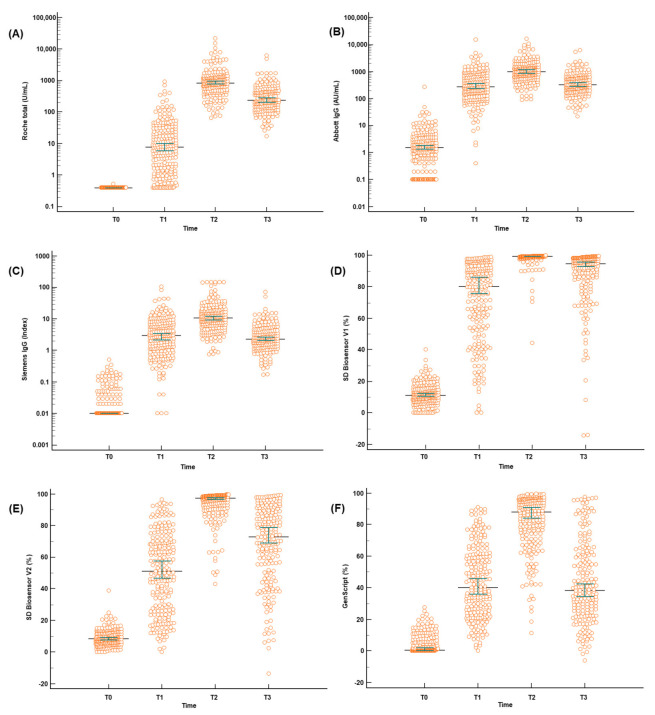
Quantitative serological response after a ChAdOx1 nCoV-19 vaccination: (**A**) Roche total antibody; (**B**) Abbott IgG; (**C**) Siemens IgG; (**D**) SD Biosensor V1 neutralizing antibody; (**E**) SD Biosensor V2 neutralizing antibody; and (**F**) GenScript neutralizing antibody. Antibody titers from T0 (before injection), T1 (1 month), T2 (4 months), and T3 (7 months) are illustrated. The differences between T2 and T3 of all the included assays were significant (*p* < 0.001). *p*-values were calculated using the Steel-Dwass-Critchlow-Finger test for quantitative differences. A logarithmic scale was applied to the y axes of the plots for Roche total, Abbott IgG, and Siemens IgG.

**Table 1 diagnostics-12-00085-t001:** Positivity values of severe acute respiratory syndrome coronavirus 2 (SARS-CoV-2) antibody assays at 7 months after a ChAdOx1 nCoV-19 vaccination according to the characteristics of participants.

Characteristics	Roche Total (%)	Abbott IgG (%)	*p*-Value	Siemens IgG (%)	*p*-Value	SD Biosensor (%)	*p*-Value	GenScript (%)	*p*-Value
Before vaccination (*n* = 228)	0 (0.0)	1 (0.4)		0 (0.0)		2 (0.9)		0 (0.0)	
After the 1st vaccination (*n* = 228)	193 (84.6)	211 (92.5)		172 (75.4)		206 (90.7)		151 (66.2)	
After the 2nd vaccination (*n* = 218)	218 (100.0)	218 (100.0)		214 (98.2)		218 (100.0)		214 (98.2)	
After 7 months from the 1st vaccination (*n* = 200)	200 (100.0)	194 (97.0)		172 (86.0)		196 (98.0)		132 (66.0)	
Sex			0.240		0.232		0.378		0.123
Male (*n* = 32)	32 (100.0)	30 (93.8)		30 (93.8)		32 (100.0)		25 (78.1)	
Female (*n* = 168)	168 (100.0)	164 (97.6)		142 (86.1)		164 (97.6)		107 (64.1)	
Age			0.930		0.391		0.864		0.261
21–30 (*n* = 80)	80 (100.0)	78 (97.5)		67 (85.9)		78 (97.5)		48 (60.8)	
31–40 (*n* = 46)	46 (100.0)	44 (95.7)		42 (93.3)		45 (97.8)		36 (78.3)	
41–50 (*n* = 44)	44 (100.0)	43 (97.7)		36 (81.8)		43 (97.7)		29 (65.9)	
51–60 (*n* = 30)	30 (100.0)	29 (96.7)		27 (90.0)		30 (100.0)		19 (63.3)	
Occupation			0.899		0.650		0.532		0.508
Doctor (*n* = 14)	14 (100.0)	14 (100.0)		13 (92.9)		14 (100.0)		9 (64.3)	
Nurse (*n* = 130)	3 (100.0)	126 (96.9)		109 (85.2)		126 (96.9)		82 (63.6)	
Laboratory technician (*n* = 54)	54 (100.0)	52 (96.3)		49 (90.7)		54 (100.0)		39 (72.2)	
Others (*n* = 2)	2 (100.0)	2 (100.0)		1 (100.0)		2 (100.0)		2 (100.0)	

Positive rates are expressed as number (%). *p*-values for Roche total could not be calculated because of 100.0% of positivity.

**Table 2 diagnostics-12-00085-t002:** Quantitative serological responses after a ChAdOx1 nCoV-19 vaccination according to the sampling time after vaccination.

Assays	T0 (*n* = 228)	Difference between T0 and T1 (*n* = 228)	T1 (*n* = 228)	Difference between T1 and T2 (*n* = 218)	T2 (*n* = 218)	Difference between T2 and T3 (*n* = 200)	T3 (*n* = 200)	*p*-Value
Roche total (U/mL)	<0.4	7.6 (1.3–26.3)	8.0 (1.7–26.7)	840.7 (470.0–1256.9)	860.5 (485.3–1286.5)	570.5 (314.0–902.8)	232.0 (138.0–469.8)	<0.001
Abbott IgG (AU/mL)	1.5 (0.5–3.3)	274.8 (113.1–723.7)	278.4 (114.3–732.8)	617.1 (168.1–1359.2)	1041.5 (631.5–1681.9)	658.9 (365.7–1103.6)	325.5 (184.8–599.7)	<0.001
Siemens IgG (index)	0.01 (0.01–0.02)	2.9 (1.0–7.1)	3.0 (1.0–7.1)	6.2 (1.8–14.0)	10.9 (6.2–17.7)	8.4 (4.4–14.0)	2.3 (1.3–4.3)	<0.001
SD Biosensor V1 (%)	11.2 (7.5–15.4)	69.1 (41.2–79.3)	81.1 (55.0–91.9)	15.9 (6.6-41.2)	99.5 (98.9–99.6)	4.8 (1.4–14.0)	94.7 (85.2–98.2)	<0.001
SD Biosensor V2 (%)	8.3 (5.4–11.9)	45.3 (16.4–62.3)	52.0 (27.8–69.9)	37.7 (20.6–64.6)	97.4 (93.0–99.0)	24.3 (9.3–39.3)	72.8 (51.3-88.7)	<0.001
GenScript (%)	0.7 (0.1–7.5)	37.7 (21.3–56.2)	40.6 (24.4–59.5)	37.5 (20.1–59.8)	88.5 (74.2–95.4)	40.5 (25.1–51.3)	38.2 (24.0–61.4)	<0.001

Data are expressed as medians (the 1st to 3rd quartiles). *p*-values between T2 and T3. T0, before the injection; T1, 1 month after the first vaccination; T2, 4 months after the first vaccination and 1 month after the second vaccination; T3, 7 months after the first vaccination.

**Table 3 diagnostics-12-00085-t003:** Agreement rates between the five SARS-CoV-2 assays ^1^.

A/B	P/P (*n*)	P/N (*n*)	N/P (*n*)	N/N (*n*)	Positive Agreement of A to B (%)	Negative Agreement of A to B (%)	Positive Agreement of B to A (%)	Negative Agreement of B to A (%)	Total Agreement (%)	Kappa Value
Roche total/Abbott IgG	602	9	22	239	96.5 (94.7–97.8)	96.4 (93.2–98.3)	98.5 (97.2–99.3)	91.6 (87.5–94.6)	96.4 (95.0–97.6)	0.91 (0.88–0.94)
Roche total/Siemens IgG	553	55	5	256	99.1 (97.9–99.7)	82.3 (77.6–86.4)	91.0 (88.4–93.1)	98.1 (95.6–99.4)	93.1 (91.2–94.7)	0.84 (0.81–0.88)
Roche total/SD biosensor	602	8	20	241	96.8 (95.1–98.0)	96.8 (93.8–98.6)	98.7 (97.4–99.4)	92.3 (88.4-95.3)	96.8 (95.4–97.9)	0.92 (0.89–0.95)
Roche total/GenScript	494	116	3	258	99.4 (98.2–99.9)	69.0 (64.0–73.6)	81.0 (77.6–84.0)	98.9 (96.7–99.8)	86.3 (83.9–88.6)	0.71 (0.66–0.76)
Abbott IgG/Siemens IgG	621	64	1	241	99.8 (99.1–100.0)	79.0 (74.0–83.4)	90.7 (88.2–92.7)	99.6 (97.7–100.0)	93.0 (91.2–94.5)	0.83 (0.79–0.87)
Abbott IgG/SD biosensor	623	8	7	241	98.9 (97.7–99.6)	96.8 (93.8–98.6)	98.7 (97.5–99.5)	97.2 (94.3–98.9)	98.3 (97.2–99.0)	0.96 (0.94–0.98)
Abbott IgG/GenScript	623	126	0	248	100.0 (99.4–100.0)	66.3 (61.3–71.1)	83.2 (80.3–85.8)	100.0 (98.5–100.0)	87.4 (85.1–89.4)	0.69 (0.64–0.74)
Siemens IgG/SD biosensor	556	1	63	248	89.8 (87.2–92.1)	99.6 (97.8–100.0)	99.8 (99.0–100.0)	79.7 (74.8–84.1)	92.6 (90.7–94.3)	0.83 (0.79–0.87)
Siemens IgG/GenScript	484	74	11	300	97.8 (96.1–98.9)	80.2 (75.8–84.1)	86.7 (83.6–89.4)	96.5 (93.8–98.2)	90.2 (88.0–92.1)	0.80 (0.76–0.84)
SD biosensor/GenScript	496	125	0	249	100.0 (99.3–100.0)	66.6 (61.5–71.3)	79.9 (76.5–83.0)	100.0 (98.5–100.0)	85.6 (83.1–87.9)	0.69 (0.65–0.74)

Agreement rates are expressed as % (95% confidence interval). N, negative; P, positive; *n*, number.

**Table 4 diagnostics-12-00085-t004:** Correlations among the five SARS-CoV-2 antibody assays according to the sampling time after vaccination.

Compared assays	After the First Vaccination (*n* = 228)	After the Second Vaccination (*n* = 218)	After 7 Months from the First Vaccination (*n* = 200)	Total	*p*-Value
Roche total/Abbott IgG	0.808	0.919	0.907	0.884	<0.001
Roche total/Siemens IgG	0.803	0.933	0.917	0.876	<0.001
Roche total/SD Biosensor V1 nAb	0.781	0.568	0.857	0.928	<0.001
Roche total/GenScript nAb	0.803	0.840	0.790	0.878	<0.001
Abbott IgG/Siemens IgG	0.947	0.975	0.980	0.976	<0.001
Abbott IgG/GenScript nAb	0.848	0.868	0.817	0.924	<0.001
Abbott IgG/SD Biosensor V1 nAb	0.854	0.579	0.846	0.905	<0.001
Siemens IgG/SD Biosensor V1 nAb	0.845	0.594	0.861	0.906	<0.001
Siemens IgG/GenScript nAb	0.845	0.871	0.819	0.932	<0.001
SD Biosensor V1 nAb/GenScript nAb	0.905	0.628	0.892	0.928	<0.001

Correlation data are expressed as Spearman’s coefficient of the rank correlation (rho). nAb, neutralizing antibody.

**Table 5 diagnostics-12-00085-t005:** Summary of studies for the longetivity of the SARS-CoV-2 antibodies after ChAdOx1 nCoV-19 and mRNA vaccinations.

Vaccine	Number of Participants	Sampling Time	Measurement	Results	Reference
ChAdOx1 nCoV-19	200	1, 4, and 7 months after the 1st dose	Roche total Abbott IgG Siemens IgG SD Biosensor nAb GenScript nAb	Roche (U/mL): 8.0→860.5→232.0 Abbott (AU/mL): 278.4→1041.5→325.5 Siemens (index): 3.0→10.9→2.3 SD Biosensor V1 (%): 81.1→99.5→94.7 GenScript (%): 40.6→88.5→38.2	This study
BNT162b2	258	1.5 and 3 months after the 2nd dose	Architect IgG	Abbott (AU/mL): 9356→3952	[6]
BNT162b2	200	14, 28, 42, 56, and 90 days after the 1st dose	Roche total	Roche (U/mL): 38.2→2204→1863→1517→1262	[7]
BNT162b2	283	8, 22, 36, 50 days after the 1st dose 111 days after the 2nd dose	Roche IgG GenScript nAb	Roche (U/mL): <20→2304→1504→761 GenScript nAb (%): 14.3→53.8→97.2→96.3→92.7	[8]
BNT162b2	517	69 days after the 2nd dose	BIO-SHIELD IgG	BIO-SHIELD (ratio): 4.23	[5]
mRNA-1273	33	209 days after the 1st dose	RBD ELISA Pseudovirus neutralization Live-virus neutralization	Maximum: 36 to 43 days ELISA (GMT): 92,451 Pseudovirus neutralization (ID_50_): 80 Live-virus neutralization (ID_50_): 406	[9]
mRNA-1273	201	16, 42, 86, and 174 days after the 1st dose	LIAISON IgG	LIAISON (AU/mL): 90.8→>400→>400→221	[10]
mRNA-1273	8	29, 43, 119, and 209 days after the 1st dose	Pseudovirus neutralization Live-virus neutralization ACE2 competition Cell-surface spike binding S-2P-binding assay RBD-binding assay	Maximum: 43 days (% of detectable antibodies) Pseudovirus neutralization: 25→100→100→88 Live-virus neutralization: 83→100→100→100 ACE2 competition: 100→100→100→100 Cell-surface spike binding: 100→100→100→100 S-2P-binding assay: 100→100→100→100 RBD-binding assay: 100→100→100→100	[11]
mRNA-1273	34	119 days after the 1st dose	RBD ELISA Pseudovirus neutralization Live-virus neutralization	Maximum: 36 to 43 days ELISA (GMT): 235,228 Pseudovirus neutralization (ID_50_): 182 Live-virus neutralization (ID_50_): 775	[12]

ACE2, angiotensin-converting enzyme 2; GMT, geometric mean end-point titers; ID_50_, 50% inhibitory dilution; nAb, neutralizing antibody; RBD, receptor-binding domain; S-2P, stabilized soluble spike protein.

## Data Availability

Jeong S, Lee N, Kim HS. 2021. Dataset of serological responses of 874 samples from healthcare workers using five SARS-CoV-2 antibody assays. Deposited in https://dataverse.harvard.edu/ (doi.org/10.7910/DVN/BXTUIR accessed on 30 December 2021).

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
