# Peer review of "Seven-Month Analysis of Five SARS-CoV-2 Antibody Assay Results after ChAdOx1 nCoV-19 Vaccination: Significant Decrease in SARS-CoV-2 Antibody Titer"

_diagnostics, 2021, doi:10.3390/diagnostics12010085_

Round 1

Reviewer 1 Report

Authors compare results from 5 different antibody assays on serum samples collected over a period of 7 months from more than 200 healthcare workers that received ChAdOx1 nCoV-19 vaccine. 

The report is one of the first describing the longevity of antibody response to ChAdOx1 nCoV-19 vaccine. Results from different assays are consistent with each other, indicating a gradual decay of humoral response over time. 

Nevetheless, the authors base their work on the assumption that antibody titer can also be measured using neutralization tests (purchased by GenScript and SD Biosensor), which are assays that evaluate the inhibition of ACE2-RBD binding by antibodies raised against viral Spike. Whether antibody titer and neutralizing activity are strictly associated is still a matter of debate (please see  doi: 10.1002/jmv.26605 and doi: 10.1002/jmv.27287 and ), despite only neutralizing antibodies are considered a reliable correlate of protection.

I would suggest to revise the manuscript considering this point. 

Minor comments :

  • Storage conditions of serum samples should be indicated
  • log scale on y axis should be indicated in figure1
  • more generally, experimental procedures should be described more in details.

Reviewer 2 Report

In the present work, Jeong and collaborators compared the serum levels of antibody against SARS-CoV-2 in response to ChAdOx1 nCoV-19 vaccination at four different time points. Using five different antibody assays, they generated data from a large cohort of health-care workers over a period of seven months that encompass the administrations of the two doses of ChAdOx1 nCoV-19 vaccine. Interestingly, they report that the positive rates measured at the last time point (T3) were decreased compared to the second dose administration (T2) and that the median values of antibody titers were significantly decreased at the same two time points. Overall, this is a well-designed experimental work, which provides a solid support to estimate ChAdOx1 nCoV-19 vaccine long-term efficacy. Nonetheless, I think some points need to be addressed:

Major points

  • I think more details regarding the different antibody assays need to be included. Standard curves and negative and positive controls should also be provided to compare the overall background and efficiency of each assay.

  • The authors should expand their discussion and conclusions on the adequacy and efficiency of each assay compared to the others. Based on their data, is one assay more performing or more suitable than other for evaluating the SARS-CoV-2 antibody titer in response to ChAdOx1 nCoV-19 vaccine? I think the authors should speculate more on this topic.

  • In paragraph 3.2 and table 1 in the result section, it is unclear whether the 4 subjects that resulted negative at T2 assessment are the same in the Siemens and GenScript assays, as well as whether they have been considered for the T3 in the 200 subjects evaluated. If so, 172 positive subjects for the Siemens assay and 132 positive subjects for GenScript assay should give 87,75% (172/196) instead of 86% (172/200) and 67,34% (132/196) instead of 66% (132/200) respectively.

Minor point.

  • In paragraph 3.3 and table 2 in the result section, I would suggest to also report the median value of the delta of antibody variation for each assay.

  • Please correct lines 149 and 151 “mRNA127” in mRNA1273.

Round 2

Reviewer 1 Report

The authors have properly addressed by comments.

Thank you

Reviewer 2 Report

The authors have nicely addressed my points.